Comparative transcriptome analysis of cold-tolerant and -sensitive asparagus bean under chilling stress and recovery

Miao Mingjun 1 2
Tan Huaqiang 3
Liang Le 1
Huang Haitao 4
Chang Wei 2
Zhang Jianwei 1
Li Ju 2
Tang Yi 1
Li Zhi 2
Lai Yunsong 1
Yang Liang 2
Li Huanxiu huanxiuli62@163.com 1
1 College of Horticulture, Sichuan Agricultural University , Chengdu , Sichuan , China
2 Horticulture Research Institute, Sichuan Academy of Agricultural Sciences , Chengdu , Sichuan , China
3 Chengdu Academy of Agriculture and Forestry Sciences , Chengdu , Sichuan , China
4 Mianyang Academy of Agricultural Sciences , Mianyang , Sichuan , China
Uversky Vladimir
Electronic publication date: 2022 Mar 22
Publication date: 2022
Volume: 10
Electronic Location ID: e13167
Received 2021 Oct 20; Accepted 2022 Mar 4
Copyright: ©2022 Miao et al.
Copyright year: 2022
Copyright holder: Miao et al.
License: This is an open access article distributed under the terms of the Creative Commons Attribution License, which permits unrestricted use, distribution, reproduction and adaptation in any medium and for any purpose provided that it is properly attributed. For attribution, the original author(s), title, publication source (PeerJ) and either DOI or URL of the article must be cited.
License URL: https://creativecommons.org/licenses/by/4.0/

Keywords: Asparagus bean, Sesquipedialis, RNA-seq, Cold stress, Recovery, Transcriptome

Funding: The Experts of Sichuan Vegetable Innovation Team-Breeding of New Vegetable Varieties of Sichuan province No. sccxtd-05, 2019-2023 This work was supported by the Experts of Sichuan Vegetable Innovation Team-Breeding of New Vegetable Varieties of Sichuan province (No. sccxtd-05, 2019-2023). The funders had no role in study design, data collection and analysis, decision to publish, or preparation of the manuscript.

==============================
Background

Low temperature is a type of abiotic stress that threatens the growth and yield of asparagus bean. However, the key genes and regulatory pathways involved in low temperature response in this legume are still poorly understood. Methodology. The present study analyzed the transcriptome of seedlings from two asparagus bean cultivars—Dubai bean and Ningjiang 3—using Illumina RNA sequencing (RNA-seq). Correlations between samples were determined by calculating Pearson correlation coefficients (PCC) and principal component analysis (PCA). Differentially expressed genes (DEGs) between two samples were identified using the DESeq package. Transcription factors (TF) prediction, Gene Ontology (GO) and Kyoto Encyclopedia of Genes and Genomes (KEGG) enrichment analysis of DEGs were also performed.

Results

Phenotypes and physiological indices indicated that Ningjiang 3 seedlings tolerated cold better than Dubai bean seedlings, in contrast to adult stage. The transcriptome dynamics of the two cultivars were closely compared using Illumina RNA-seq following 0, 3, 12, and 24 h of cold stress at 5 °C and recovery for 3 h at 25 °C room temperature. Global gene expression patterns displayed relatively high correlation between the two cultivars (>0.88), decreasing to 0.79 and 0.81, respectively, at 12 and 24 h of recovery, consistent with the results of principal component analysis. The major transcription factor families identified from differentially expressed genes between the two cultivars included bHLH, NAC, C2H2, MYB, WRKY, and AP2/ERF. The representative GO enrichment terms were protein phosphorylation, photosynthesis, oxidation-reduction process, and cellular glucan metabolic process. Moreover, KEGG analysis of DEGs within each cultivar revealed 36 transcription factors enriched in Dubai bean and Ningjiang 3 seedlings under cold stress.

Conclusions

These results reveal new information that will improve our understanding of the molecular mechanisms underlying the cold stress response of asparagus bean and provide genetic resources for breeding cold-tolerant asparagus bean cultivars.

Introduction

Low temperature is an essential environmental factor that limits plant distribution, growth, development, and yield (Yang et al., 2020). According to temperature ranges and corresponding physiological changes in plants, low-temperature stress comprises chilling (0 °C–15 °C) and freezing (<0 °C) (Megha, Basu & Kav, 2014). Plants originating from tropical and subtropical climatic zones are often threatened by chilling stress, which challenges the yield and quality of economically important crops (Suh et al., 2010; Xu et al., 2008). Chilling stress causes changes in membrane fluidity and cytoskeleton rearrangement, followed by an influx of calcium through calcium channels into the cytoplasm, triggering downstream responses, including changes in ion transport, metabolism, post-translational protein modifications, and gene expression (Guo, Liu & Chong, 2018).

The transcriptome is the complete set of transcripts in a cell, both in terms of type and quantity, for a specific developmental stage or physiological condition (Wang, Gerstein & Snyder, 2009). To characterize and investigate the transcriptome, many technologies have been developed, including hybridization-based microarrays and Sanger sequencing-based methods (Yamada et al., 2003; Cheung et al., 2008; Fu, Chen & Hsiao, 2011). With the advent of next generation sequencing (NGS), RNA sequencing (RNA-seq) has become a novel technology for transcriptome analysis and is a powerful and effective tool for identifying differentially expressed genes (DEGs) and probable molecular mechanisms (Vogel et al., 2016; Wei et al., 2017; Yang et al., 2019; Liu et al., 2019). RNA-seq is mainly based on a next generation sequencing (NGS platform) with new and complete library building, sequencing, and analysis systems (Meyers et al., 2004). The transcriptomes of cold-tolerant and cold-sensitive plants under low-temperature stress have been compared using RNA-seq over the past decade (Liu et al., 2012; Liu et al., 2019; Yang et al., 2020).

Cowpea (Vigna unguiculata L. Walp.) is one of the most important grain legumes worldwide and is cultivated in the warm to hot regions of Africa, Asia, and the Americas (Ehlers & Hall, 1997). As a warm-season legume crop, cowpea can adapt to drought and heat, but it is sensitive to chilled temperatures (Hall, 2004; Agbicodo et al., 2009; Ismail, Hall & Close, 1999). Asparagus bean (Vigna unguiculata subsp. sesquipedalis) is a subspecies of cowpea that is mainly grown in eastern and southern Asia to produce immature green pods (Xu et al., 2011a; Xu et al., 2011b). Low temperatures during early spring and late autumn negatively affect the normal growth of these plants at the seedling and pod set stages, and eventually lead to a significant loss in yield and quality.

Current investigations into the low-temperature tolerance of cowpea are still mainly physiological, whereas knowledge of related molecular mechanisms is scant. Ismail et al. found that the specific dehydrin (DHN) 1 in the seeds of cowpea (subsp. unguiculata) conferred cold tolerance during germination (Ismail, Hall & Close, 1999; Ismail, Hall & Close, 1997). Zuo et al. (2018) combined RNA- and sRNA-seq to analyze the miRNA and mRNA expression profiles and regulatory networks of cowpea pods under chilling stress. Tan et al. (2016) explored differences in gene expression between the asparagus bean cultivars, Dubai bean and Ningjiang 3 with different degrees of cold tolerance using RNA-seq under 0 and 24 h of chilling stress. However, the experimental design was relatively simple, having only two time points.

The aim of this study was to determine the transcriptome dynamics of Dubai bean and Ningjiang 3 during cold stress and recovery at room temperature (seven time points in total) using Illumina RNA-seq and uncover the genes and mechanisms of different asparagus bean genotypes associated with cold stress. Our results provide updated information and genetic resources for improving the low-temperature tolerance of asparagus bean.

Materials & Methods

Plant material

We selected the asparagus bean cultivars Dubai bean and Ningjiang 3 that are respectively cold-tolerant and -sensitive to chilling temperatures when adults (Huang et al., 2018). Seeds were soaked at 25 °C for 4 h and then sown in watered plugs containing vermiculite and perlite (V:V = 1:1). The plugs were placed in a growth chamber at 25 °C under 14 h light/10 h dark cycles with light intensity of 30,000 lx and watered every two days after seedling emergence. After 2 weeks, growth chamber temperature was reduced to 5 ° C, and light intensity was set to 6,000 lx. Seedling leaves were harvested 0, 3, 12, and 24 h later. Growth conditions were returned to 25 °C and 30,000 lx and seedling leaves were harvested at 3, 12, and 24 h later. Three biological replicates of each cultivar were sampled at each of seven time points.

Cultivar phenotyping

Half of the leaf samples were frozen in liquid nitrogen and stored at −80 °C. Fresh leaves were weighed; proline and superoxide dismutase (SOD) contents in the remaining samples were evaluated using respective assay kits (Cat. Nos. A107-1-1 and A101-1-1; Nanjing Jiancheng Bioengineering Institute, Nanjing, China) as described by the manufacturer.

Extraction of RNA and library construction

Total RNA was extracted using the RNAiso Plus kit (Takara Bio Inc., Kusatsu, Japan) as described by the manufacturer. The integrity and quality of the RNA was confirmed by 1.5% agarose gel electrophoresis and using a NanoDrop ND-1000 spectrophotometer (Thermo Fisher Scientific Inc., Waltham, MA, USA).

Since 21 RNA samples were extracted from each cultivar, we constructed a total of 42 sequencing libraries using NEBNext® Ultra™ RNA Library Prep Kit for Illumina (New England Biolabs Inc., Ipswich, MA, USA) as described by the manufacturer. Libraries were sequenced on an Illumina Hiseq X platform, and 125–150 bp paired-end reads were generated.

Data analysis

Raw sequencing reads were filtered by trimming adapters and removing low quality reads. Clean reads were mapped to the cowpea genome using HISAT2 software (Kim, Langmead & Salzberg, 2015; Lonardi et al., 2019), and reads mapped to each gene were counted using HTSeq v. 0.6.1. The fragments per kilobase of transcript per million mapped reads (FPKM) of each gene was calculated according to their length and the number of reads mapped to them (Anders, Theodor & Huber, 2015). Correlations between samples were determined using Pearson correlation coefficients (PCC). Principal component analysis (PCA) was performed using Factoextra and FactoMineR package in R software.

Differentially expressed genes (DEGs) between Dubai bean and Ningjiang 3 under cold stress and recovery were analyzed using the DESeq package in R (ver. 1.18.0). Resulting P values were adjusted using the Benjamini and Hochberg approach to control false discovery rates (FDRs). Genes with adjusted P < 0.05, and log2 fold change ≥ 2 were considered differentially expressed.

Gene Ontology (GO) enrichment of DEG sets was analyzed using the BiNGO plugin of Cytoscape (Maere, Heymens & Kuiper, 2005). Transcription factors (TF) were identified in DEGs using iTAK (Zheng et al., 2016). Pathways of gene sets were analyzed using MapMan v. 3.6.0RC1 tool. Enriched Kyoto Encyclopedia of Genes and Genomes (KEGG) enrichment was analyzed using TBtools (Chen et al., 2020).

Quantitative RT-PCR analysis

We validated the RNA-seq results using 15 cold-responsive genes, including 10 AP2/ERF transcription factors, four circadian rhythm-related genes, and ICE1. Table S1 shows the primers that were designed using Primer3 (http://bioinfo.ut.ee/primer3-0.4.0/) and synthesized (Invitrogen; Thermo Fisher Scientific Inc.). Quantitative RT-PCR was performed using three technical replicates and SYBR Premix Ex Taq TM II (Takara Bio Inc.) on a Bio-Rad CFX96 Real-Time PCR system (Bio-Rad Laboratories Inc., Hercules, CA, USA). The PCR amplification conditions were 95 °C for 3 min, followed by 40 cycles of 95 °C for 10 s and 60 °C for 30 s. The relative expression of the selected genes was normalized to that of the cowpea actin gene (GenBank ID: LOC114185033) and calculated using the 2−ΔΔCT method.

Results

Cultivar phenotyping under chilling stress and recovery

One symptom of chilling injury is a water deficit in shoots due to imbalanced water transport and transpiration (Kang & Saltveit, 2001). Under cold stress at 5 °C, the leaves of both Dubai bean and Ningjiang 3 gradually drooped and wilted (Fig. 1). After 3 h of recovery at 25 °C, the leaves of both varieties continued to droop. At 12 and 24 h of recovery, the leaves of Dubai bean remained drooped and crinkled, whereas most Ningjiang 3 leaves not only stopped wilting, but also showed signs of resumed growth.

Figure 1 Phenotypes of Dubai bean and Ningjiang 3 seedlings under cold stress and recovery.

C0, C3, C12, C24, cold stress at 5 ° C for 0, 3, 12, 24 h, respectively. R3, R12, R24, recovery at 25 ° C for 3, 12, 24 h.

Changes in leaf fresh weight and proline and SOD contents

The fresh weight of Dubai bean and Ningjiang 3 leaves progressively decreased under cold stress at 5 °C (Fig. 2). After 3 h of recovery at 25 °C, the leaf fresh weight of Dubai bean and Ningjiang 3 was unchanged and slightly decreased, respectively. At recovery for 12 and 24 h, the fresh weight of Dubai bean leaves continued to decline, whereas that of Ningjiang 3 increased and was only 10% lower than in cold stress for 0 h (Table S2).

Figure 2 Leaf fresh weight and proline and SOD contents of leaf samples from Dubai bean and Ningjiang 3 under cold stress and recovery stages.

* P < 0.05 and ** p < 0.01 (one-way ANOVA).

Antioxidant enzymes, such as SOD, catalase (CAT), and ascorbate peroxidase (APX), help plants clear excessive reactive oxygen species (ROS) produced at low temperatures (Yu et al., 2006). In addition, compatible solutes, such as proline and carbohydrates, play important roles in cell osmotic adjustment and maintaining membrane integrity under cold stress (Liu et al., 2012). We examined proline and SOD contents after cold stress at 5 °C and after recovery for 3 h at 25 °C (Fig. 2). We found a gradual increase in the proline and SOD content of leaves from both plants, but these values declined or remained unchanged after recovery for 12 and 24 h (Tables S3 and S4). However, proline content at C3, R12 and R24 time points and SOD content at C3, C12, C24 and R3 time points were significantly higher in Ningjiang 3 than in Dubai bean (P < 0.01).

Taken together, we initially inferred that cold tolerance is better for Ningjiang 3 than Dubai bean at the seedling stage based on the phenotype, leaf fresh weight and proline and SOD contents.

Global transcriptome results for Dubai bean and Ningjiang 3

We used Illumina RNA-seq to investigate transcriptome dynamics at seven time points during cold stress and recovery using total RNA isolated from leaves of Dubai bean and Ningjiang 3. Three independent biological replicates of leaves at each time point were analyzed (21 samples each in total).

Over 2 billion high-quality reads (average ∼49 million reads per sample) were generated from each plant at different time points (Table S5) and mapped to the cowpea genome (Lonardi et al., 2019) at a mean rate of 93.72% (range: 89.41%–96.06%). The R package RNASeqPower (Hart et al., 2013) was used to calculate the statistical power of this experiment. A power of 0.65 was found for genes that presented fold-change greater than 2 with a sample size of three (the number of biological replicates used in this study).

The reads mapped to each gene were counted using HTSeq v. 0.6.1, and the FPKM of each gene was calculated (Anders, Theodor & Huber, 2015). Genes with FPKM >1 were considered to be expressed. At seven time points of cold stress and recovery, 16,211–17,961 and 16,174–17,297 genes were respectively expressed in Dubai bean and Ningjiang 3 (Fig. 3A). The expression of 10%–12% of genes was very high (FPKM ≥ 50) at different time points (Fig. 3B). The expression of genes with high (FPKM 10 to ≥ 50), moderate (FPKM 2 to ≤ 10), and low (FPKM 0.1 to ≤ 2), respectively was 27%–34%, 25%–29%, and 28%–34%. For genes showing high/very high expression, more genes were expressed in Dubai bean during cold stress and in Ningjiang 3 during recovery.

Figure 3 Genes expressed in Dubai bean and Ningjiang 3 during cold stress and recovery.

(A) All genes expressed in two varieties over seven time-points under cold stress and during recovery. (B) Fraction of genes with differential expression in two varieties over seven time-points under cold stress and during recovery. D, Duban bean; N, Ningjiang 3.

Comparison of global transcriptome between two cultivars

We investigated global differences in the Dubai bean and Ningjiang 3 transcriptomes during cold stress and recovery. We calculated Pearson correlation coefficients (PCC) of all samples from seven time points based on the mean FPKM of three biological repeats for all expressed genes (Fig. 4A). The PCC values between the two cultivars under cold stress at 5 °C for 0, 3, 12, and 24 h, were 0.93, 0.96, 0.97, and 0.88, respectively, and 0.94, 0.79, and 0.81 at 25 °C recovery for 3, 12, and 24 h, respectively. The correlation between samples of the two cultivars under 5 °C and recovery at 25 °C for 3 h was close (PCC >0.88) but not at 12 h of recovery at 25 °C (PCC = 0.79).

Figure 4 Correlations between the transcriptomes of different stages of cold stress and recovery in Dubai bean and Ningjiang 3 cultivars.

(A) Pearson correlation coefficient (PCC) analysis of RNA-seq data from seven stages of cold stress and recovery in Dubai bean and Ningjiang 3. (B) Principal component analysis (PCA) of samples at different stages of cold stress and recovery in two asparagus bean varieties. Blue triangles, Ningjiang 3; red dots, Dubai bean; ellipses, 95% confidence intervals.

We conducted PCA using the factoextra and FactoMineR packages in R (Fig. 4B). Samples with closer distances suggested more similar transcriptional programs. Samples under cold stress at 5 °C were distributed on the left side of the plot, while samples under recovery at 25 °C were distributed on the right, indicating that the low-temperature response and recovery at room temperature are different processes. The cultivars under cold stress 5 °C and after 3 h of recovery at 25 °C were closely grouped, but more distanced at 12 h of recovery at 25 °C (Fig. 4A). The results indicated that the two cultivars are more likely to have similar transcriptomes and functions/activities under cold stress at 5 °C and recovery for 3 h at 25 °C. In contrast, transcriptional activity at 12 h of recovery was somewhat distinct and might explain the difference in cold tolerance between the two asparagus bean cultivars.

Comparison of DEGs between Dubai bean and Ningjiang 3

Figure 5A shows the upregulated expression of 18,257 genes (including 745 TF-encoding genes) and the downregulated expression of 15,142 genes (including 572 TF-encoding genes) at different stages of cold stress and recovery in Dubai bean compared with Ningjiang 3. Most (8,954 and 7,455) genes were differentially expressed after recovery for 12 (R12), and 24 (R24) hours, whereas the minimum (1,947) was differentially expressed after cold stress for 3 h (C3).

Figure 5 Differential gene expression between Dubai bean and Ningjiang 3 at various stages of cold stress and recovery.

(A) Differentially expressed genes in Dubai bean vs. Ningjiang 3 at all stages. Blue, green and purple indicate upregulated genes, downregulated genes, and transcription factors (TFs), respectively. (B) Ratios of upregulated and downregulated genes expressing different TFs in Dubai bean during cold stress and recovery.

Transcription factor prediction of DEGs between Dubai bean and Ningjiang 3

Transcription factors (TFs) play a vital role in regulating biotic and abiotic stress response networks (Liu et al., 2019). We investigated differentially expressed TFs to better understand the molecular mechanisms of the low-temperature responses in the two varieties (Fig. 5B). The number of DEGs in some families differed between the two cultivars (Table S6). For instance, more TFs (including bHLH, NAC, WRKY, bZIP, and AP2/ERF) were upregulated in Dubai bean, whereas most downregulated TFs in Dubai bean were in the C2H2, Aux/IAA, TCP, HB-KNOX, and DDT families. Transcription factors in the zf-HD and FAR1 families were upregulated in Dubai bean and Ningjiang 3, respectively.

GO enrichment of DEGs between Dubai bean and Ningjiang 3

We investigated the potential functions of DEGs by analyzing GO enrichment (Fig. 6). A few GO terms were enriched under cold stress, such as responses to auxin and microtubule-based movement. However, more terms, such as protein phosphorylation, photosynthesis, oxidation–reduction process, and cellular glucan metabolic process were enriched during recovery, suggesting their functional participation in cold tolerance of asparagus bean.

Figure 6 Gene Ontology (GO) terms enriched with downregulated and upregulated differentially expressed genes (biological process) at various stages of cold stress in Dubai bean compared with Ningjiang 3.

Colored scale, corrected P values.

Comparison of DEG pathways between Dubai bean compared with Ningjiang 3

We classified the functions of the DEGs using Mercator4 v. 3.0 to further determine functional groups (Lohse et al., 2014). The log2 fold changes of DEGs between Dubai bean and Ningjiang 3 at each stage were loaded into the Overview pathway using the MapMan tool, and the functions of DEGs were classified (Fig. 7A). The top five groups with most DEGs were RNA biosynthesis, protein modification, solute transport, cell cycle organization, and cell wall organization. The ratios of upregulated genes in Dubai bean were essentially stable in the first three groups, fluctuated in cell wall organization, and gradually declined in cell cycle organization.

Figure 7 Functional classifications and DEG pathways in Dubai bean compared with Ningjiang 3.

(A) Functional groups of DEGs at seven cold stress and recovery stages and their expression. Dot sizes and colors indicate number and ratios of upregulated genes, respectively, in Dubai bean compared with Ningjiang 3. (B) Metabolic pathways with differential expression in Dubai bean vs. Ningjiang 3 after recovery for 12 h. Red and blue indicate upregulated and downregulated genes, respectively, in Dubai bean. DEG, differentially expressed gene.

The ratios of upregulated genes associated with photosynthesis in Dubai bean under cold stress cold stress were >65% at 0, 3, 12 h and 7% at 24 h, 83% within 3 h of recovery, then respectively decreased to 20% and 5% at 12 and 24 h of recovery. The trend was the same for coenzyme metabolism. Some other pathways including chromatin organization, RNA processing, DNA damage response, protein biosynthesis, and protein homeostasis, gradually declined, indicating that these functions were adversely affected in Dubai bean at low temperatures.

We investigated the metabolic pathways responsible for the differences in cold tolerance between Dubai bean and Ningjiang 3 at 12 h of recovery. We loaded the log2 fold changes in DEGs onto the X4.1 Metabolism overview R1.0 pathway using MapMan (Fig. 7B), because the correlation between the two varieties was the lowest (PCC = 0.79) at this stage. The differential activity of metabolic pathways at 12 h of recovery was consistent with the PCA results. Most of the genes involved in photosynthesis, including those associated with photophosphorylation and the Calvin cycle, were downregulated in Dubai bean, indicating impaired photosynthetic function. In addition, most genes involved in cell wall synthesis and modification, amino acid synthesis, and nucleotide metabolism were also downregulated in Dubai bean. These results indicated that the disrupted cell functions of Dubai bean had not yet returned to normal despite 12 h of recovery under normal temperatures.

Identification of DEGs in Dubai bean and Ningjiang 3

We analyzed DEGs in each cultivar at different stages of cold stress and recovery by plotting Venn diagrams using Venny2.1 (Fig. 8). Compared with 0 h, 3773, 6286, and 7362 genes in Dubai bean were differentially expressed after cold exposure for 3, 12, 24 h, respectively. We identified 2,026 DEGs that were common to all three stages. Furthermore, 2,374, 2,894, and 5,527 DEGs were identified in Ningjiang 3 exposed to 5 °C for 3, 12, 24 h, respectively, and 1,636 genes were common to all three stages. We identified 7,005 and 5,994 DEGs that were common to Dubai bean and Ningjiang 3, respectively, under all three recovery stages.

Figure 8 Venn diagrams of differentially expressed genes in Dubai bean and Ningjiang 3 during cold exposure and recovery.

D, Dubai bean; N, Ningjiang 3. C0, C3, C12 and C24, cold exposure for 0, 3, 12 and 24 h, respectively. R3, R12 and R24, recovery at room temperature for 3, 12 and 24 h, respectively.

KEGG analysis of DEG enrichment in Dubai bean and Ningjiang 3

We blasted common DEGs into the KEGG database to identify the primary biological pathways in which these common DEGs were involved (Fig. 9). Enriched pathways unique to Dubai bean during cold exposure included chromosome and associated proteins, enzymes with EC numbers, and glycosylphosphatidylinositol (GPI)-anchored proteins. Those in Ningjiang 3 included plant hormone signal transduction, pentose and glucuronate interconversions, and porphyrin metabolism. Enriched KEGG pathways common to both cultivars included the MAPK signaling pathway, plant-pathogen interaction, and transcription factors. Ribosome biogenesis, photosynthesis, chaperones and folding catalysts, and glyoxylate and dicarboxylate metabolism were enriched KEGG pathways in both cultivars during recovery. These annotations provide a basis for understanding the common and different biological pathways involved in the two asparagus bean cultivars under cold stress and recovery.

Figure 9 KEGG pathway enrichment findings of common differentially expressed genes (DEGs) in Dubai bean and Ningjiang 3 under cold stress and recovery.

DEGs common to Dubai bean and Ningjiang 3 under cold stress (CDC and CNC, respectively) and recovery at room temperature (CDR and CNR, respectively). Colored scale, P values.

Enriched transcription factors in Dubai bean and Ningjiang 3

The results of the KEGG pathway enrichment analysis revealed TFs that were enriched in DEGs that were common of both cultivars under cold stress, indicating their involvement in the low-temperature responses of asparagus bean. We identified 36 differentially expressed TFs in this pathway (Table 1); 28 and 25 were in Dubai bean and Ningjiang 3, respectively. These TFs belonged to the MYB (12), AP2/ERF (11), WRKY (6), HSF (3), HD-ZIP (2), E2F/DP (1), and NF-YB (1) families. Homologous genes of TFs, comprising ERF (LOC114194502), MYB (LOC114169018 and LOC114168556), and three WRKY (LOC114194094, LOC114184078 and LOC114188798) are involved in the cold response of Arabidopsis, according to annotations in The Arabidopsis Information Resource (TAIR) database.

Table 1 Enriched transcription factors by KEGG enrichment analysis in Dubai bean and Ningjiang 3.

Gene ID	TF family	Hits in Arabidopsis	NC24/NC0	NR24/NC24	DC24/DC0	DR24/DC24	
LOC114176637	AP2	AT3G20840.1	0.52	6.68	−1.4	4.93	
LOC114193917	AP2	AT1G16060.1	−1.38	−0.81	−1.34	−0.48	
LOC114176859	E2F/DP	AT3G48160.2	0.04	4.91	−1.43	0.18	
LOC114194502	ERF	AT1G78080.1	2.95	−2.6	2.08	−1.14	
LOC114177032	ERF	AT5G25810.1	0.86	−1.22	1.09	−3.18	
LOC114191951	ERF	AT3G16770.1	2.71	−2.96	5.43	−4.18	
LOC114190586	ERF	AT3G23240.1	5.61	−1.83	4.72	−3.7	
LOC114167378	ERF	AT4G17500.1	4.64	−2.58	4.46	−2.49	
LOC114177898	ERF	AT2G31230.1	2.7	0.17	1.74	−1.52	
LOC114173447	ERF	AT4G17490.1	3.22	−2.56	3.69	-5	
LOC114168274	ERF	AT5G44210.1	4.23	−3.06	3.26	−2.35	
LOC114186996	ERF	AT1G50640.1	−2.37	1.16	−2.69	1.44	
LOC114168763	HD-ZIP	AT1G05230.4	−0.44	0.7	−1.74	−0.17	
LOC114184235	HD-ZIP	AT3G60390.1	−1.18	−0.62	1.41	−3.11	
LOC114193441	HSF	AT1G46264.1	−4.19	4.57	−1.33	1.66	
LOC114193339	HSF	AT2G26150.1	−3.62	3.66	−2.64	3.33	
LOC114192670	HSF	AT3G22830.1	−3.58	2.44	−3.97	0.88	
LOC114162527	MYB	AT1G17950.1	0.17	−0.73	1.98	−0.13	
LOC114190831	MYB	AT2G47460.1	−1.16	0.46	−1.21	−1.99	
LOC114163641	MYB	AT5G49330.1	−2.17	−0.52	−1.67	−1.94	
LOC114173857	MYB	AT5G57620.1	1.99	1.72	1.42	−1.04	
LOC114183416	MYB	AT2G37630.1	−0.83	0.35	−1.44	−0.81	
LOC114169018	MYB	AT3G23250.1	6.13	−1.08	4.48	−4.59	
LOC114181503	MYB	AT3G28910.1	2.21	−2.54	2.54	−4.04	
LOC114181399	MYB	AT5G15310.1	−1.38	0.14	−1.95	−0.47	
LOC114194518	MYB	AT4G37260.1	2.84	−2.11	2.86	−1.11	
LOC114168556	MYB	AT3G23250.1	10.71	−7.8	13.87	−7.05	
LOC114189637	MYB	AT1G09540.1	2.02	−4.37	0.54	−3.08	
LOC114170581	MYB	AT3G46130.1	−2.66	1.08	−1.53	0.97	
LOC114193521	NF-YB	AT4G14540.1	−1.81	0.46	−1.86	−0.57	
LOC114167396	WRKY	AT2G03340.1	6.23	−3.99	5.31	−6.42	
LOC114194094	WRKY	AT2G38470.1	4.55	−1.83	3.5	−1.75	
LOC114184078	WRKY	AT2G38470.1	4.89	−2.59	4.32	−4.65	
LOC114177110	WRKY	AT4G23550.1	2.86	3.49	2.28	1.66	
LOC114190737	WRKY	AT5G52830.1	−1.41	−0.02	1.54	−2.8	
LOC114188798	WRKY	AT2G30250.1	3.88	0.23	3.03	0.32	
Notes.

Numbers in last four columns mean Log2FoldChange of FPKM value.

Validation by qRT-PCR

We validated the accuracy and reliability of RNA-seq data using 15 cold-responsive genes comprising 10 AP2/ERF genes encoding TFs, four associated with the circadian clock, and ICE1 (Xu et al., 2011a; Xu et al., 2011b; Dong, Farré & Thomashow, 2011; Chinnusamy et al., 2003). The expression of these genes determined by qRT-PCR was similar to those determined using RNA-seq (Fig. S1). The mean correlation coefficient between the RNA-seq and real-time PCR results was 0.83 (0.66–0.97), indicating that the RNA-seq data accurately reflected the abundance of transcripts (Tables S7 and S8).

Discussion

Asparagus bean is a warm-season legume that can tolerate heat, but not cold. The low-temperature response of asparagus bean at the physiological level has been investigated (Ismail & Hall, 2002; Li et al., 2005; Chen, Wang & Ding, 2005). However, the ultimate effects of low-temperature stress on plant growth depend not only on degrees of damage, but also on the capacity for recovery thereafter (Zhang & Scheller, 2004). Chilling stress and recovery in asparagus bean has not been systematically investigated as far as we can ascertain. Therefore, this study analyzed responses of the asparagus bean cultivars Dubai bean and Ningjiang 3 with different degrees of tolerance to low temperature stress and recovery at room temperature. The phenotype and physiological indices showed that Ningjiang 3 tolerated cold better than Dubai bean at the seedling stage. In contrast, adult Dubai bean tolerates cold stress better than adult Ningjiang 3 (Huang et al., 2018). These findings indicated that the molecular mechanisms at the seedling and adult stages may differ between the two cultivars. Whether genes that confer chilling tolerance at one developmental stage affect that at other developmental stages has remained unknown (El-kholy, Hall & Mohsen, 1997). Tomato cultivars that function well under chilling stress during germination do not always function well at fruit set (Kemp, 1968). The chilling tolerance of two cucumber genotypes did not differ between the fruit and germination stages (Jennings & Saltveit, 1994). Ye et al. (2009) evaluated the low-temperature response of 17 rice cultivars at various growth stages. They found that the same cold tolerance at all four growth stages in seven cultivars, and slight differences among growth stages in six. The cold tolerance of the other four cultivars considerably differed during growth stages, especially particularly the Amaroo and Azucena cultivars that are cold tolerant as seedlings but cold sensitive at the flowering stage. Our results were similar to these findings. Therefore, some cultivars even within in the same species might have different degrees of cold tolerance at specific growth stages. This might be due to growth conditions in the place of origin, as well as breeding history (Ye et al., 2009). Although cold tolerance is easier to rate at the seedling stage than at the adult stage, tolerance at these stages might be irrelevant or even contradictory. Hence, to assume that a cold-tolerant cultivar at the seedling stage is also resistant to low temperatures at the adult stage and can be cultivated during the late autumn may be unreasonable.

We calculated PCC between the Dubai bean and Ningjiang 3 at different stages based on the average FPKM of three biological replicates. The results revealed a closer correlation between samples of the two cultivars exposed to cold stress at 5 °C and recovery for 3 h at 25 °C (PCC >0.88). The correlation was lower at 12 and 24 h of recovery (PCC 0.79 and 0.81, respectively). The results of a subsequent PCA analysis were consistent, indicating that the two cultivars have similar physiological and molecular responses to stress at 5 °C. However, at 12 and 24 h of recovery at 25 °C, the phenotypes and transcriptomes of the two cultivars considerably differed, indicating that the difference in cold tolerance between them was mainly reflected during recovery. Both cold-tolerant and cold-sensitive rice and maize varieties are impaired under low temperature, but cold-tolerant varieties recover better (Saruyama & Tanida, 1995; Ribas-Carbo et al., 2000; Yu et al., 2006). A temperature of 5 °C might be below the threshold for both Dubai bean and Ningjiang 3 to withstand cold stress. Hence, the phenotypes of these cultivars were similar. However, Ningjiang 3 contained more SOD and proline than Dubai bean. The results of other crops are similar (Guo et al., 2006; Jahnke, Hull & Long, 1991; Khan et al., 2019; Tian et al., 2021; Taïbi et al., 2018).

High activities of ROS-scavenging enzymes are associated with resistance to low-temperature stress (Yu et al., 2006; Apel & Hirt, 2004). Superoxide dismutase acts as the first line of defense against ROS by dismutating superoxide to H2O2 and reducing the risk of hydroxyl radical formation from superoxide (Zhang et al., 2011). Proline accumulation can help to maintain the osmotic balance of cells, reduce water loss, and improve the adaptability of crops to low-temperature environments (Taïbi et al., 2018). Ningjiang 3 seemed to gradually resume growth during recovery, but not Dubai bean. This might be because Ningjiang 3 has a higher osmolyte content and more powerful antioxidant enzyme activity, thus reducing the damage caused by low-temperature stress on its growth.

The representative TF families identified from DEGs during cold stress and recovery between two asparagus been varieties included bHLH, NAC, C2H2, MYB, WRKY and AP2/ERF, which was similar to previous findings (Yang et al., 2020; Tan et al., 2016; Liu et al., 2019). These TF families are also associated with cold tolerance in other plants (Xu et al., 2011a; Xu et al., 2011b; Chinnusamy et al., 2003; Zhai et al., 2010; Shao, Wang & Tang, 2015; Tripathi, Rabara & Rushton, 2014; Kim et al., 2001). Therefore, these TF families might also participate in the cold tolerance of asparagus bean. Our KEGG enrichment findings of DEGs within each variety showed that TFs in the MYB, AP2/ERF, and WRKY families were enriched (Table 1). Homologous genes of six TFs, including one ERF (LOC114194502), two MYB (LOC114169018 and LOC114168556), and three WRKY (LOC114194094, LOC114184078, and LOC114188798) in Arabidopsis, are involved in the cold response. LOC114194502 (AT1G78080.1) encodes RAP2.4, which belongs to the DREB subfamily A-6 of the AP2/ERF TF family. Levels of RAP2.4 mRNA are obviously elevated upon exposure to cold and high salinity (Rae, Lao & Kavanagh, 2011). Both LOC114169018 and LOC114168556 (AT3G23250.1) encode AtMYB15, an R2R3 type MYB transcription factor that participates in the cold regulation of CBF genes and the development of cold tolerance. The expression of MYB15 is upregulated under cold stress, and MYB15 protein interacts with ICE1 and binds to MYB recognition sequences in CBF gene promoters (Hao et al., 2006). Both LOC114194094 and LOC114184078 (AT2G38470.1) encode AtWRKY33, and LOC114188798 (AT2G30250.1) encodes AtWRKY25. The abundance of transcripts of these genes increases after exposure to cold stress for 24 h (Jiang & Deyholos, 2009). The present study also found upregulated expression of the six genes encoding TFs under cold stress described above. However, this decreased after recovery for 24 h at room temperature. The results showed that these TFs are involved in the common responses of two asparagus bean cultivars to cold temperatures.

Chilled temperatures significantly impact plant photosynthesis. All major components of photosynthesis, including thylakoid electron transport, the carbon reduction cycle, and control of stomatal conductance, can be disrupted (Allen & Ort, 2001). Exposing Arabidopsis to cold stress at 5 °C for 3 days inhibited light-saturation rates of CO2 assimilation and decreased the yield of PSII electron transport and PSI activity (Savitch et al., 2001). The photosynthetic traits of tomato plants, including net photosynthetic rate, stomatal conductance, internal CO2 concentration, and transpiration rate, significantly decrease upon exposure to low temperature stress (Khan et al., 2019). The photosynthetic rates of soybean seedlings are also substantially reduced after cold stress for 12 h at 5 °C (Tian et al., 2015). The gross photosynthesis of Vigna species, including cowpea, is susceptible to chilling (Briiggemann, 1992). Our KEGG findings of DEGs revealed that photosynthesis-related pathways were commonly enriched in Dubai bean and Ningjiang 3 during recovery at room temperature. These results indicated that the photosynthesis processes of both varieties were affected during recovery. The GO findings of DEGs showed that genes related to photosynthesis were downregulated after 24 h at 5 °C as well as after 12 and 24 h of recovery at room temperature in Dubai bean compared with Ningjiang 3; These results were consistent with the findings determined using MapMan (Fig. 7A). In addition, the metabolic pathways of DEGs analyzed at 12 h of recovery showed that most of the photosynthesis-related genes in Dubai bean, including photophosphorylation and the Calvin cycle, were downregulated compared with Ningjiang 3. These results were consistent with those of other studies. Liu et al. (2019) investigated the differences in physiology and global gene expression between cold-tolerant and cold-sensitive tomato genotypes under cold stress. All tomato genotypes showed a time-dependent decline in Fv/Fm and ΦPSII, but the decrease was greater in the sensitive genotype. Analysis of DEGs enriched in GO terms showed that cold stress significantly inhibited photosynthesis, and that more downregulated genes in the cold-sensitive genotype were involved in all aspects of photosynthesis, including light reactions, the Calvin cycle, and photorespiration. These results suggest that photosynthesis is more suppressed in the cold-sensitive, than the cold-tolerant genotype under cold stress. Moreover, although low temperatures inhibit photosynthesis in cold-sensitive and cold-tolerant genotypes in some crops such as Musa sp., Zea mays, and Miscanthus, the cold tolerant genotype functioned better (Zhang et al., 2011; Ting, Owens & Wolfe, 1991; Friesen & Sage, 2016). These results indicated that the photosynthesis functions were more damaged in Dubai bean than Ningjiang 3 during cold stress and recovery.

Conclusions

We analyzed transcriptomes of the asparagus bean cultivars, Dubai bean and Ningjiang 3, with different degrees of cold tolerance after exposure to stress at 5 °C for 0, 3, 12, and 24 h followed by recovery at 3, 12, and 24 h under room temperature using Illumina RNA-seq. We also measured the physiological indices of leaf fresh weight, proline levels, and SOD contents. Ningjiang 3 was more cold-tolerant than Dubai bean at the seedling stage, which was opposite to the adult stage. This might be due to a higher osmolyte content, more powerful antioxidant enzyme activity, more rapid photosynthetic rate, and better restoration of the water balance. These results provide a systematic understanding of the mechanism underlying low temperature responses and recovery in asparagus bean and lay a foundation for the genetic improvement of cold-sensitive asparagus bean germplasm in the future.

Supplemental Information

Supplemental Information 1 Validation of expression level of selected genes in Dubai bean and Ningjaing 3

Heatmaps represent expression profiles of selected genes (labeled on the right side) obtained from RNA-seq (left) and qRT-PCR (right) analysis. The color scale at the bottom represents Z-score. The values between the two heatmaps represent the correlation between expression profiles of selected genes obtained from RNA-seq and qRT-PCR analysis.

Click here for additional data file.

Supplemental Information 2 Primers used for qRT-PCR validation of selected genes

Click here for additional data file.

Supplemental Information 3 Raw data of leaf fresh weight of Dubai bean and Ningjiang 3 under cold stress and recovery (6 repeats)

C0, C3, C12, C24 represent 5 ° C cold stress for 0, 3, 12, 24 h, respectively. R3, R12, R24 indicate 25 ° C recovery for 3, 12, 24 h, respectively.

Click here for additional data file.

Supplemental Information 4 Raw data of proline content of Dubai bean and Ningjiang 3 under cold stress and recovery (3 repeats)

C0, C3, C12, C24 represent 5 ° C cold stress for 0, 3, 12, 24 h, respectively. R3, R12, R24 indicate 25 ° C recovery for 3, 12, 24 h, respectively.

Click here for additional data file.

Supplemental Information 5 Raw data of SOD content of Dubai bean and Ningjiang 3 under cold stress and recovery (3 repeats)

C0, C3, C12, C24 represent 5 ° C cold stress for 0, 3, 12, 24 h, respectively. R3, R12, R24 indicate 25 ° C recovery for 3, 12, 24 h, respectively.

Click here for additional data file.

Supplemental Information 6 Alignment result of illumina reads to reference genome

N stands for Ningjiang 3 and D stands for Dubai bean. C0, C3, C12, C24 represent 5 ° C cold stress for 0, 3, 12, 24 h, respectively. R3, R12, R24 indicate 25 ° C recovery for 3, 12, 24 h, respectively.

Click here for additional data file.

Supplemental Information 7 Statistics of transcription factors upregulated in Ningjiang 3 and Dubai bean at all stages

N stands for Ningjiang 3 and D stands for Dubai bean. C0, C3, C12, C24 represent 5 ° C cold stress for 0, 3, 12, 24 h, respectively. R3, R12, R24 indicate 25 ° C recovery for 3, 12, 24 h, respectively.

Click here for additional data file.

Supplemental Information 8 Raw data of FPKM value of selected genes calculated by RNA-seq

N stands for Ningjiang 3 and D stands for Dubai bean. C0, C3, C12, C24 represent 5 ° C cold stress for 0, 3, 12, 24 h, respectively. R3, R12, R24 indicate 25 ° C recovery for 3, 12, 24 h, respectively.

Click here for additional data file.

Supplemental Information 9 Raw data of relative expression value of selected genes calculated by qPCR

N stands for Ningjiang 3 and D stands for Dubai bean. C0, C3, C12, C24 represent 5 ° C cold stress for 0, 3, 12, 24 h, respectively. R3, R12, R24 indicate 25 ° C recovery for 3, 12, 24 h, respectively.

Click here for additional data file.

We would like to thank Editage for English language editing.

Additional Information and Declarations

Competing Interests

Author Contributions

DNA Deposition

Data Availability

The authors declare there are no competing interests.

Mingjun Miao and Huanxiu Li conceived and designed the experiments, prepared figures and/or tables, and approved the final draft.

Huaqiang Tan and Le Liang conceived and designed the experiments, authored or reviewed drafts of the paper, and approved the final draft.

Haitao Huang, Wei Chang, Jianwei Zhang and Ju Li performed the experiments, authored or reviewed drafts of the paper, and approved the final draft.

Yi Tang, Zhi Li, Yunsong Lai and Liang Yang analyzed the data, prepared figures and/or tables, and approved the final draft.

The following information was supplied regarding the deposition of DNA sequences:

The raw sequencing data generated in this study are available at the Sequence Read Archive (SRA) of the National Center for Biotechnology Information (NCBI): PRJNA763044.

The following information was supplied regarding data availability:

The data is available at NCBI SRA: SRR15882990–SRR15883031.

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
