# Peer review of "Comparative transcriptome analysis of cold-tolerant and -sensitive asparagus bean under chilling stress and recovery"

_PeerJ, doi:10.7717/peerj.13167_

## Round 0.1 · original submission · Major Revisions

As you can see, reviewers raised multiple concerns. Therefore, major revision is needed. Please carefully address all the critiques of the reviewers.

Reviewer 1 ·

Basic reporting

In this study, authors have employed RNA sequencing to investigate the transcriptome dynamics of Dubai and Ningjiang 3 beans during cold stress and normal temperature recovery. The results presented show Ningjiang 3 beans have higher cold tolerance than Dubai beans at the seedling stage.

Authors should use clear and unambiguous English language across the manuscript. One such example is Abstract line 22-23 "However, the key genes and regulatory pathways that respond to low temperature in this legume are still poorly understood". These lines should be reframed. Authors should conduct thorough proofreading of the manuscript.

Line 90/Line 113 Authors should provide full forms of all the abbreviations used at the first instance of their occurrence in the manuscript for the general audience of the journal. For example SOD, DEGs

Line 180 "For genes…. stages" extra "High" should be dropped.

Line 317-319 "After low…….. Variety". This paragraph is poorly written making it hard for the reader to follow.

Experimental design

See comments below.

Validity of the findings

Line 138-140 "However, at 12h and 24h …. growth". It is not evident from Figure 1 (R12, R24 images) that there is any visual difference between the leaves of the two Cultivars. Authors should provide additional evidence supporting the claims made in lines 138-140.

Figure 2 Authors should elaborate on why the leaf fresh weight is lower for Dubai beans whereas the SOD and proline content is matched between the two bean varieties at time 0?

Figure 3 A Is there a statistically significant difference in the number of genes expressed by Dubai beans compared to Ningjiang 3 beans for the R24 time point? Line 175 of the results section authors claimed the number to be higher but there is no statistical evidence provided for the same.

Line 223-225 "Besides…. Respectively" the statement is vague and should be rewritten for the readers to follow. Zf-HD1 is upregulated as shown in Figure 5B.

Line 317 "Some reports …. results". Please provide relevant references to support this statement.

Tan et al. employed RNA-Seq to explore the gene expression differences between the same asparagus bean varieties with different cold tolerance under 0 and 24 hrs stress. How are the findings in the current study any different from those already reported in the literature? What is the value-add of having more time points? Authors should discuss the differences/ new findings if any between their results to those previously reported in the literature.

·

Basic reporting

Writeup needs a lot of edits.
Professional English edit is needed before any acceptance can be ma

Experimental design

pathetic

Validity of the findings

Invalidated results and conclusions.

Additional comments

Introduction is trash, there is no new method description in the Introduction section, what the authors want to do ? The introduction is definitely loosely written and is important relevant work in the field.
Why the two beans were selected? What is the logic behind the selection? No reference given appropriately.
Result section is definitely incomplete and data is missing and experiments have not been well thought of.
The authors say they calculated SOD content and proline content.. but why was no experiments done to probe CAT assays and APX enzyme based assays? More background information on probing proline content should be explicitly mentioned.
This paper needs a major revamp.

·

Basic reporting

In this study, Miao et al. evaluated the cold tolerance of two asparagus bean varieties at the seedling stage by using phenotype, physiological indexes, and transcriptome comparison. They found some major transcription factor families and cold-responsive genes may play roles in the cold tolerance of Ningjiang 3. This research is interesting and provides a better understating of the cold stress mechanism for the two asparagus bean varieties. The manuscript is well written and easy to read. I recommend doing major revision.

Experimental design

1. For the cold stress treatment of two seedlings, the authors changed temperatures between 25℃/5℃and light intensity between 30,000 lx/6000 lx. And all the experiments the authors did later were based on this cold stress treatment. If the authors need to focus on the effect of cold stress on the seedling growth, why did the authors change the light intensity? If so, all the conclusions authors obtained in this study not only resulted from cold stress but also from light density stress.
2. In figure 1, the seedlings of Ningjiang3 and Dubai bean showed different growing statuses at the beginning of cold stress (C0): Ningjiang3 looks much taller and stronger than Dubai bean, which may cause Ningjiang3 to show better cold tolerance. Thus the authors should use two seedlings with similar growth statuses in height, size, and age. If it is hard to get similar seedlings, at least the author should repeat this study with taller/bigger Dubai bean and shorter/smaller Ningjiang3 to see whether the author can obtain similar results and conclusions.
3. In this study, the author mentioned “the cold tolerance of Dubai bean at adult stage is higher than that of Ningjiang 3. This indicated that the molecular mechanisms at seedling stage and adult stage were quite different between the two asparagus bean varieties (lines 284-287).” If drew the conclusion that the molecular mechanisms at the seedling stage and adult stage were quite different, the authors should provide some evidence, did the authors check the changes of transcription factor genes and cold-responsive genes by using the qRT-PCR experiments in Figure 8 for the two adult asparagus beans after cold stress-recovery cycle?
4. In Figure 5A and Figure 6, the authors compared the differential gene expression in Dubai bean as compared with Ningjiang 3 at different stages of cold stress and recovery. For example, in Figure 5A, DC0 vs NC0, there were 3487 gene increased and 1741 genes decreased in Dubai compared to Ningjiang3, but those gene differences are not due to cold stress, similarly, not all the gene changes for DC3 vs NC3, DC12 vs NC12, and other bars come from cold stress. If the authors want to find gene changes affected by cold stress, why not do comparisons with Dubai bean for DC0 vs DC3, DC0 vs DC12, and DC0 vs DC24, DC24 vs DR3, DC24 vs DR12, and DC24 vs DR24, as well as do a similar comparison for Ningjiang3 seedlings?

Validity of the findings

Please do statistical analysis for Figure 2.

---

## Round 0.2 · Major Revisions

Please address the remaining concerns of the reviewer and revise the manuscript accordingly.

·

Basic reporting

The authors have addressed most of my concerns in the revised version, but I am still worried about the much big different sizes of the two seedlings used in this study, which may cause the conclusions the authors claimed in this study to be inaccurate.

Experimental design

For most models used in research, we generally allow reasonable ranges for ages, size, weight, etc. Unfortunately, the seedlings of two different varieties used in this study were obviously out of a reasonable range of size. There are several methods to fix the defect of different seedling sizes, for example, the authors can verify and confirm their transcriptome data findings in two different seedlings with similar sizes. Actually, this verification is quite important and necessary for transcriptome data, however, the authors did not try to address the issue.

Validity of the findings

The authors need to verify and confirm their transcriptome data findings in two different seedlings with similar sizes.

---

## Round 0.3 · accepted · Accept

All remaining concerns were adequately addressed and the manuscript was amended accordingly. Therefore, the revised version is acceptable in its present form.